# Glomerular Macrophages in Human Auto- and Allo-Immune Nephritis

**DOI:** 10.3390/cells10030603

**Published:** 2021-03-09

**Authors:** Solange Moll, Andrea Angeletti, Leonardo Scapozza, Andrea Cavalli, Gian Marco Ghiggeri, Marco Prunotto

**Affiliations:** 1Department of Pathology, University Hospital of Geneva, 1205 Geneva, Switzerland; Solange.Moll@hcuge.ch; 2Nephrology, Dialysis and Transplantation Unit, Giannina Gaslini Scientific Institute for Research, Hospitalization and Healthcare, 16147 Genoa, Italy; andreaangeletti@gaslini.org (A.A.); GMarcoGhiggeri@gaslini.org (G.M.G.); 3Institute of Pharmaceutical Sciences of Western Switzerland, School of Pharmaceutical Sciences, University of Geneva, 1205 Geneva, Switzerland; Leonardo.Scapozza@unige.ch; 4Institute for Research in Biomedicine (IRB), Università della Svizzera Italiana (USI), 6900 Lugano, Switzerland; andrea.cavalli@irb.usi.ch; 5Galapagos NV, 4051 Basel, Switzerland

**Keywords:** glomerulonephritis, macrophages, immune nephritis

## Abstract

Macrophages are involved in tissue homeostasis. They participate in inflammatory episodes and are involved in tissue repair. Macrophages are characterized by a phenotypic heterogeneity and a profound cell plasticity. In the kidney, and more particularly within glomeruli, macrophages are thought to play a maintenance role that is potentially critical for preserving a normal glomerular structure. Literature on the glomerular macrophage role in human crescentic glomerulonephritis and renal transplantation rejection with glomerulitis, is sparse. Evidence from preclinical models indicates that macrophages profoundly modulate disease progression, both in terms of number—where depletion has resulted in a reduced glomerular lesion—and sub-phenotype—M1 being more profoundly detrimental than M2. This evidence is corroborated by better outcomes in patients with a lower number of glomerular macrophages. However, due to the very limited biopsy sample size, the type and role of macrophage subpopulations involved in human proliferative lesions is more difficult to precisely define and synthesize. Therefore, specific biomarkers of macrophage activation may enhance our ability to assess their role, potentially enabling improved monitoring of drug activity and ultimately allowing the development of novel therapeutic strategies to target these elusive cellular players.

## 1. Introduction

Macrophages (Mφs) are a group of circulating/resident mononuclear cells involved in tissue homeostasis, characterized by a phenotypic heterogeneity and a profound cell plasticity. They participate in inflammatory episodes and are involved in tissue repair [1]. Together with monocytes, osteoclasts and dendritic cells (DCs), Mφs are essential components of the phagocytic system and play a pivotal role in the response to infectious, toxic, ischemic, autoimmune or inflammatory stimuli; in doing so, they constitute the link between innate and adaptive immunity [2,3].

All mononuclear cells originate from the hematopoietic system, subsequently migrating to reach different peripheral organs [2,4] and conferring their own phagocytic system on those tissues. Upon stimuli, the resident cell pool is expanded, releasing chemokines that in turn recruit additional monocytes from circulation via a specialized process named diapedesis. This results in the crossing of the vascular endothelium and the sub-endothelial matrix [5,6].

Data on the composition of the human renal phagocytic system in normal conditions are extremely sparse and essentially rely on either the analysis of “normal” tissue areas distal to a cancerous lesion obtained from routine nephrectomies [7], or on biopsy areas without lesions, derived from patients with non-identifiable diseases. Based on those studies, the normal human kidney mainly possesses a DC system composed of interstitial DCs and DC precursors. Although up to date human knowledge remains limited, in the last two decades preclinical studies in both autoimmune models and in models of immune-mediated glomerulonephritis have allowed the scientific community to present near conclusive considerations on the pathologic implications of macrophages in the kidney (see specific sections).

This article reviews the literature available on the role of macrophages in both glomerulonephritis and in vivo renal transplantation, covering an almost forty-year period from the early 80 s to present day. The focus is limited to the glomerulus: a specific area of the kidney specialized in blood filtration and the target of immune processes active in inflammatory, autoimmune and alloimmune conditions.

Two preliminary sections, dedicated to the analysis of macrophage heterogeneity and their phenotypic specificity in the renal compartments, were included to facilitate understanding of the role of macrophage in glomerulonephritis and renal transplantation.

## 2. Macrophage Heterogeneity and Functions

The monocytic cell lineage includes three cell types that all derive from bone marrow [8] and differentiate, upon specific stimuli, into DCs, M1 or M2 Mφs (Figure 1); differentiated monocytes play protective functions in normal tissues and participate individually in pathological processes [9]. As previously described, selected monocytic cell types reside in the tissues where, despite significant functional overlap, they can be recognized based on their phenotypic specificities. In response to signals released locally, and directly related to infections such as pathogen associated molecular patterns (PAMPs) or derived from injured tissues (DAMPs) [10], resident cells proliferate in a time period that can last seconds or hours. In the local microenvironment, the different components of the monocytic cell lineage play different roles: DCs assure immune surveillance and modulate adaptive immunity, whereas Mφs, through their phagocytic activity, participate in both inflammation and tissue repair [11,12].

DCs play a main role as antigen presenting cells (APCs), acting as a red thread between innate and adaptive immunity, internalizing and processing antigens, and presenting peptides bound to the major histocompatibility complexes I and II to T cells [3,13]; overall they shape the local immunity for peripheral tolerance [14]. Conventionally defined by their CD11c positivity, the quantity of DCs in circulation is low, representing less than 0.1% of total circulating leukocytes. They primarily reside within the normal renal interstitium, and only rarely in glomeruli.

Mφs encompass a heterogeneous group of cells devoted to phagocytosis of cell debris derived from cell apoptosis [15]. Two main phenotypes exist, defined largely by in vitro stimulation: M1, or pro-inflammatory Mφs, and M2, or wound healing Mφs. Similarly to DCs, Mφs originate from the bone marrow and proliferate and differentiate upon stimulation by specific cytokines, such as the macrophage colony-stimulating factor (M-CSF). In the normal kidney, Mφs represent a minimal quota of resident monocytic cells and can rapidly infiltrate both the tubule-interstitial and glomerular compartments in pathologic conditions. Mφs are conventionally characterized by double positivity for the myeloid lysosomal membrane protein CD68 and the cell surface myeloid cell CD11b. Polarization into M1 Mφs occurs upon exposure to cytokines such as interferon-γ, or to pathogen endotoxins (PAMPs) or DAMPS, with a correspondent increased expression of Ly6C [16] and CD38 [17] considered a typical marker of the pro-inflammatory activated M1 Mφ phenotype. Activated M1 Mφs generate inflammatory cytokines (IL1β, IL6, NOS2, ROS), growth factors (FGF2), angiogenic (VEGF) substances, and signaling proteins involved in tissue differentiation (WNT7B) [15,18]. Conversely, exposure to cytokines such as IL10 and IL4 induces polarization of Mφs toward M2 with the acquisition of an anti-inflammatory phenotype; polarized M2 are prototypically characterized by the expression of the scavenger receptor markers CD206, CD86 and CD163 and are involved in tissue repair and fibrosis. In the early 2000s, an extended classification was proposed in which M2 Mφs, which are immune cells with high phenotypic heterogeneity, were classified into subgroups (M2a, b, c and d) according to their exposure to different stimuli and to their achieved transcriptional changes [19,20,21,22]. M2a Mφs are activated by IL-4 or IL-13, express the surface marker CD206 and lead to the increased expression of TNF-α, IL-1a, IL-1β, IL-6, and IL-12 and IL-23 (PMID: *31037072*). M2b Mφs are activated by immune complex, Toll-like receptor (TLR) ligands and IL-1β, express the surface marker CD86 and release both pro- and anti-inflammatory cytokines, such as TNF-α, IL-1β, IL-6, and IL-10. M2c Mφs, also known as inactivated Mφs, are induced by glucocorticoids, IL-10 and TGF-β, express the surface marker CD163 and secrete IL-10, TGF-β, CCL16, and CCL18. M2d Mφs are induced by IL6 and adenosines, express the surface marker IL-10R and lead to the release of IL-10 and vascular endothelial growth factors (VEGF) [21,22,23,24]. Today, M2 Mφs are identified based on their expression of a set of markers which are transmembrane glycoproteins, scavenger receptors, enzymes, growth factors, hormones, cytokines, and cytokine receptors with diverse and often yet unexplored functions [21]. In the 2010s, a systematic transcriptome-based network analysis of human Mφs exposed to a wide range of signals has further extended the spectrum of activation states well beyond the original M1/M2 dichotomy [25]. Moreover, single cell analysis in pathology has further amplified the diversity of functional states of Mφss under physiological and pathological conditions [26]. It is important to note that Mφs are exposed to a multiplicity of signals in vivo with different temporal patterns [19]. Therefore, polarization of Mφs should be viewed as an operationally useful, simplified conceptual framework, describing a continuum of diverse functional states [19]. Thus, M2 Mφs have high functional heterogeneity and it is still not well understood whether this heterogeneity is a result of their reversible adoption of M2 activation in response to the tissue environment or due to irreversible differentiation programs [21]. It should however be noted that these in vitro classifications do not necessarily reflect their true phenotypes in vivo. In fact, Mφs are believed to represent a spectrum of activated phenotypes rather than stable subpopulations [27]. Thus, Anders and Ryu proposed to refine the M1/M2 Mφ-working model in view of the putative Mφ phenotypes that occur during the different phases of kidney disease, namely pro-inflammatory, anti-inflammatory, pro-fibrotic and fibrolytic Mφs [28]. Thus, a switch from M1 to M2 Mφs is thought to occur during natural resolution of inflammation. Overall, Mφs respond to severe injuries by attempting lesion sterilization (via production of pro-inflammatory cytokines) and debridement (via phagocytosis), followed by a phase where they actively contribute to tissue renewal (pro-fibrotic and fibrolytic phases). In the context of a maladaptive repair, all those Mφ functions, which are essential in the context of correct tissue repair, could contribute to lesion perpetuation and disarrangement of local tissue morphology. Thus, like M1 Mφs, excessive or uncontrolled M2 Mφs activity can also cause diseases such as fibrosis.

## 3. Phenotype Overlap and Renal Compartmentation

In the kidney, a clear-cut distinction between DCs and Mφs, based on the previously cited definition (CD11c^+^ for DCs and CD68^+^/CD11b^+^ for Mφs) is not always possible [7,29]. For instance, in the inflamed kidney, CD11c^+^ DCs may also acquire cell surface markers characteristic of M2 such as CD206 [30]. By overlooking the presence of cells simultaneously co-expressing Mφ and DC markers, several studies have confused our understanding of the Mφ role–pro- or anti-inflammatory–in kidney diseases [30]. Moreover, although the original classification in M1 and M2 Mφs was helpful in early investigations, further studies have demonstrated the plasticity of Mφ and the overlap of M1 and M2 gene expression in response to complex activation signals [31].

In glomerulonephritis, a handful of studies analyzed additional monocytic cell lineage markers, in particular DC-SIGN, BDCA-2 and Langherin, typically expressed by Langherans cells. These studies show that those markers are co-expressed by different monocytic cell phenotypes [7,32]. Interestingly, Segrerer et al. [7], using confocal microscopy and double staining for CD68 and DC-SIGN, a marker expressed by myeloid dendritic cells [33,34] were able to show that CD68^+^ cells infiltrating the tubule-interstitial were DC-SIGN^+^, whereas those infiltrating the glomeruli were negative for DC-SIGN [7]. Based on that expression profile, those authors proposed the concept of phagocytic cell compartmentation for the kidney, with glomerular infiltrating CD68^+^ cells being essentially Mφs, and CD68^+^ cells infiltrating the tubule-interstitial space having essentially DC features. In vitro studies confirmed this evidence, showing that DCs–but not Mφs–express DC-SIGN [35].

In the next section, we will try to summarize what is known about glomerular Mφ phenotypes and functions in human glomerulonephritis.

## 4. Glomerular Macrophages in Human Glomerulonephritis

Most studies related to the characterization of the renal monocytic lineage cells in glomerulonephritis (GN) focused on tubule-interstitial space. This complexity has been reviewed by other authors [36,37]. In the present section, we will focus on the glomeruli, where most monocytic cells are Mφs [7,29] and only very few are DCs, as mentioned above.

Within glomeruli, Mφs are thought to play a key function, eliminating potentially dangerous antigens and clearing cell debris: a maintenance role that is potentially critical to the preservation of a normal glomerular structure. Indeed, Mφ activation is mediated by a range of stimuli including cytokines, foreign proteins and immune-complexes (ICs), comprising immunoglobulins, antigens and complement components that deposit in the normal glomeruli and characterize pathological glomeruli in case of glomerulopathies, GN with autoimmunity or GN with ICs [11]. In “non-immunological” chronic kidney diseases such as vascular or diabetic nephropathy (DN), glomerular Mφ activation is indeed also observed. Thus, the number of glomerular anti-inflammatory CD163^+^ Mφs M2 was demonstrated to be associated with pathological DN lesions, such as nodular sclerosis and global glomerulosclerosis [38]. Therefore, in these non-immunological diseases, the Mφs major function may be phagocytosis of cellular debris or pathological matrix. 

Our knowledge of Mφ function during GN is unfortunately fragmented and limited to a mere description of the presence/absence of the different monocytic cell phenotypes detected during the disease stages. Moreover, the picture is slightly complicated by the literature that, for almost a decade, from the ’90 s to 2000, relied essentially on anti-FM32 and anti-Leu 32 for monocyte characterization [39,40,41,42,43,44], followed, from 2000 onward, by a generally accepted use of CD68 as a main marker of glomerular Mφs [7,45,46]. Although –to our knowledge–the reproducibility of glomerular macrophage staining/scoring between pathologists or between laboratories has not been published, contemporary “best practices” for enumerating glomerular macrophage infiltration are based on immunohistochemistry with CD68 antibody and count number of the CD68^+^ cells using a software, as described by Wu et al. [47]. Table 1 reports all publications on that topic available in PubMed along with the total number of patients studied and Figure 2 summarizes the current available mechanistic knowledge related to Mφs in GNs.

According to the literature, we can define the following points:The number of glomerular macrophages is correlated with the severity of proliferative GN.

The first observation is that glomerular Mφs are mostly abundant in mostly proliferative GN. Thus, Mφs are detected in the glomerular tuft in almost all type of GNs. The results vary from “positive” to “intense” staining. It is accepted that the intensity of the Mφ staining, and therefore of the Mφ number, is correlated with the intensity of glomerular cellularity, namely proliferation of intrinsic glomerular cells (for example mesangial proliferation in IgA GN) or infiltration of circulating inflammatory cells (for example polymorphonuclear endocapillar infiltration in cryoglobulinemic GN). Thus, according to the concordant results of the literature, we can accept that marked intense staining is observed in proliferative GN characterized by intense glomerular cellular proliferation such as observed in crescentic GN, i.e., Lupus nephritis (LN) and ANCA associated vasculitis, whereas less intense staining is detected in less proliferative GN, i.e., IgA and membranous nephropathy [48,49,50].

2.Macrophages localize within glomeruli in the most severe lesions.

The second observation is that, in severe proliferative GN, Mφs are mostly localized in mostly severe/acute/destructive glomerular lesions. Thus, Rastaldi et al., compared glomerular CD68+Mφ localization in two severe proliferative GN; ANCA-associated GN, a disease characterized by massive glomerular Mφ infiltration with necrotizing extracapillary lesions, and cryoglobulinemic GN, a disease characterized by massive glomerular Mφinfiltration but without necrotizing extracapillary lesions [48]. The authors showed comparable Mφ numbers in glomeruli of both diseases, but differences in localization. They observed an accumulation of a great number of Mφ in glomeruli of ANCA-associated GN, especially in areas of extracapillary proliferation (crescents) and in glomerular granulomatous lesions. In contrast, in cryoglobulinemic GN, Mφ seemed to be homogeneously distributed in the glomerular tuft but not in the periglomerular interstitium.

3.Macrophages are attracted within glomeruli and glomerular lesions by chemokines produced by intrinsic glomerular cells.

The third observation concerns the type of chemoattractants and the type of intrinsic glomerular cells which play a role in attracting the macrophages. Thus, several studies highlight the presence of chemokines involved in Mφ chemoattraction in tissues [51,52]. Monocyte chemoattractant protein-1 (MCP-1) has been detected in IgA, in LN and in granulomatosis with polyangiitis (GPA, previously known as Wegener granulomatosis) [53,54], confirming what was previously reported using direct methods in their respective clinical settings. Moreover, MCP-1 and its receptor chemokine receptor 2B (CCR2B) were shown to be expressed in crescents of human crescentic GN, and CD68+ cells were demonstrated to be the main glomerular cell type that expressed CCR2B [55].

Intrinsic renal cells were identified as the major source of macrophage migration inhibitory factor (MIF) production in human GN [56]. De novo MIF expression was observed in glomerular capillary endothelium in severe disease cases with large numbers of infiltrating glomerular Mφs, suggesting that endothelial MIF production may participate in the process of glomerular Mφ recruitment. In addition, the up-regulation of MIF by glomerular parietal epithelial cells was demonstrated to largely contribute to glomerular crescent formation in human crescentic GN. Moreover, a significant correlation between renal MIF expression, degree of renal injury, degree of renal dysfunction and urine MIF concentration was demonstrated in different types of GN (IgA GN, crescentic GN and lupus GN) [57]. Finally, a significant correlation was shown between the intraglomerular and interstitial Mφ numbers and the concentration of urinary MIF [58]. It is also interesting to note that, in the same study, the levels of urinary MIF were demonstrated to have increased 2 weeks before the flaring of disease activity, reflecting the clinicopathological activity of the disease. It should be noted that it is not yet known whether distinct chemoattractants might explain the different types of glomerular lesions, however, it is likely that different types of triggers induce chemoattractant production by intrinsic renal cells leading to glomerular lesions.

4.Attracted macrophages are activated in proliferative glomerular lesions.

The fourth observation concerns the properties acquired by macrophages in cases of severe glomerular proliferative lesions. Thus, Rastaldi et al. [48], in addition to comparing glomerular CD68+ Mφ number and localization in ANCA-associated GN and cryoglobulinemic GN, analyzed their properties. They showed significant differences in adhesion, activation, cytokine production and proliferation. Indeed, adhesion, with de novo production of glomerular vascular cell adhesion molecule-1 VCAM-1, was found only in ANCA GN. It is particularly noteworthy that VCAM-1, which is fundamental in monocyte adhesion, was detected only in the areas of necrotizing extracapillary lesions, leaving the remaining tuft negative. Activation, with HLA class II and 27E10 expression, was prominent in ANCA GN. Proinflammatory cytokine production, with tumor necrosis factor-α and interleukin-1β, was prominent in ANCA GN. Moreover, TNF-α and IL-1 expression paralleled 27E10 staining, confirming in vitro results demonstrating in cell separation experiments that Mφs expressing 27E10 epitope produced the greatest quantities of TNF-α and IL-1 [59]. Finally, proliferation, with proliferative markers PCNA and Mib-1, was only observed in ANCA GN. Therefore, according to this study, ANCA GN differs profoundly from cryoglobulinemic GN in macrophage properties, giving this disease a stronger severity. It should be mentioned that similar data have been reported by the same group in other forms of glomerular capillarities, i.e., necrotizing IgA nephritis, Henoch–Schönlein syndrome, and glomerulonephritis associated with endocarditis, and never found in other glomerular diseases [60,61]. Altogether, these data suggest that acute Mφ activation directly influences the production of adhesion molecules in the endothelium and of proinflammatory cytokines in the glomeruli, making the disease more severe.

5.The difficulty of identifying macrophage subpopulations in proliferative glomerular lesions.

The fifth observation, which is more difficult to precisely define and synthesize, concerns the type and role of Mφ subpopulations involved in proliferative glomerular lesions. Indeed, in the last ten years, only a few studies were reported in the literature, most with partial subpopulation analysis (and only one which analyzed the Mφ subpopulations).

Thus, three studies analyzed CD68+/163+ immunostaining (M2c subpopulation) in biopsies with active proliferative GN obtained from patients naïve of therapy. Results were concordant: (1) Li et al., analyzed 24 IgA GN biopsies (10 with crescents, 14 without crescents) and reported significantly more CD68+/163+ Mφs in glomeruli with cellular and fibrocellular crescents, compared with glomeruli without crescents [49]. (2) The same authors confirmed their results and extended their analysis to different types of crescentic GN (22 lupus GN, 10 antineutrophil cytoplasmic autoantibody (ANCA)–associated pauci-immune necrotizing GN and 5 type 1 membranoproliferative GN) [62]. They demonstrated that CD68+/163+ M2c Mφs were mainly expressed in active crescentic glomerular lesions. (3) Finally, Zhao et al., analyzing biopsies with necrotizing GN (17 ANCA-associated pauci-immune GN, 5 anti-GBM GN, 4 SLE GN and 4 IgA GN), demonstrated that CD68+163+ M2c Mφs predominated in early stages of glomerular lesions, i.e., at sites of glomerular fibrinoid necrosis, exceeding the quantity of neutrophils and T cells [63]. These authors made another interesting observation: they observed that the most normal appearing glomeruli in ANCA-associated GN had significantly greater interstitial infiltrates of CD68+and CD163+ Mφs and PMNs than controls. This raises the question of the relationship of M2c Mφs interstitial infiltrates with the extent of glomerular disease. Indeed, M2c Mφs could be positioned as responders/anti-inflammatory effectors, as is currently accepted, or as potential effectors of glomerular injury in the very early stages of pauci-immune necrotizing GN. This question of the role of the M2c subpopulation in the pathogenesis of glomerular lesions remains open. Likewise, other questions remain unanswered and need to be addressed, since macrophages might be potential targets for therapeutic intervention: are the other subpopulations (M1, M2a and M2b) present in these initial/early glomerular lesions, is there a macrophage dedifferentiation during the evolution/progression of the glomerular lesions?

In another study using CD68+CD86+ immunostaining, the M2b subpopulation was analyzed in biopsies with active proliferative GN obtained from patients naïve of therapy [64]. Wu et al. showed in 12 ANCA and anti-GBM-associated GN that the mean numbers of intraglomerular M2b Mφs were significantly higher in crescentic glomeruli, mostly in cellular rather than in fibrocellular crescents, and particularly in crescentic glomeruli with ruptured Bowman’s capsule [64]. However, whether these M2b are present concomitantly to the M2c described in the studies above, representing separate populations, or are the same Mφs, expressing each of these markers and representing plasticity of differentiation, remains uncertain. To answer this question, studies with more global analysis of the subpopulations are needed. In this regard, two studies were reported, one on lupus GN [50], the other one on IgA GN [65].

Olmes et al. analyzed renal biopsies from 68 patients with lupus nephritis (ISN/RPS classes II–V) using immunohistochemical analysis for infiltration with M1-like (iNOS+/CD68+), M2a-like (CD206+/CD68+) and M2c-like Mφs (CD163+/CD68+) [50]. It is important to note that, among the 68 patients selected in this study, more than half (56%) were treated at the time of the biopsy with immunosuppressive drugs (85% with a combination of different medications). This point is important to take into consideration since there is some evidence to show that immunosuppressive therapy can modulate Mφ subpopulations, as suggested in vitro [66] and in transplant patients using flow cytometry analysis [67]. Concordant with other previously cited studies, the authors demonstrated a correlation between the number of glomerular Mφs and SLE ISN/RPS class. Thus, the number of total CD68+ Mφs per glomerular area was greatest in class IV renal biopsies. The major finding was that infiltration of Mφ subpopulations differed between the classes and was dominated by M2c. Thus, glomerular M2c numbers were significantly higher in class III and IV compared to class V, whereas differences among the investigated ISN/RPS classes in M1 and M2a numbers were only minor: glomerular M1 number was comparable in all classes; there were few glomerular M2a compared to M1, but when comparing classes, significantly more M2a were found only in class IV. Interestingly, the ratios of M1 to M2a and M2c, which are indicators of changes in the inflammatory milieu, were also dependent on class. Glomerular ratios of M2c/M1 were higher in class III and IV compared to class V, whereas the differences in the ratios of M2a/M1 were not significant. Glomerular M2c/M2a ratios were significantly higher in class IV compared to class V. Taken together, these results obtained in a inhomogeneous group of patients (with and without immunosuppressive therapy) are concordant with the three studies mentioned above analyzing M2c subpopulation in biopsies with active proliferative GN obtained from patients naïve of therapy. Furthermore, high concentrations of soluble CD163 released by M2c were reported in plasma from patients with SLE and correlated with the SLE disease activity index, indicating that this Mφ subtype is highly induced in active disease [68]. However, it remains unclear whether M2c Mφs are actors of disease progression, or are recruited to counteract this process in order to prevent progressive inflammation.

In contrast, another study analyzing Mφ subpopulation in IgA GN reported quite different results. Hu et al. investigated the distribution of M2 subpopulations in IgA GN and the correlation with clinicopathological features in renal biopsies obtained from 49 untreated patients [65]. M2 Mφ markers included CD206+/CD68+ (M2a), CD86+/CD68+ (M2b) and CD163+/CD68+ (M2c). M2 subpopulations were analyzed according to the different types of glomerular lesions defined by the Oxford classification of IgA nephropathy 2016 (MEST-C score with M = mesangial hypercellularity; E = endocapillary cellularity; S = segmental sclerosis; T = interstitial fibrosis/tubular atrophy; C = crescent) [69]. Concordant with results from other studies, the numbers of glomerular CD68 Mφs were higher in patients with M1, S1 and C1. In this study however, no significant trend in the glomerular subpopulation could be demonstrated according to the type of glomerular lesions:

-no differences in numbers of glomerular M2b were observed between M0 and M1, S0 and S1, T0 and T1, C0 and C1. M2b macrophages only had an increased trend in glomeruli of G1 (global glomerulosclerosis), without any significant difference.-no differences in numbers of glomerular M2c were observed between M1 and M0, S1 and S0, C1 and C0. There were fewer M2c macrophages in glomeruli with T1 and G1.

It should be noted that the total numbers of M2a Mφs were not reported in glomeruli but only in the tubule-interstitial, with larger numbers of M2a Mφs with M1, S1 and T1 but not with C1. It should also be noted that when the subpopulation was analyzed according to another old and no longer used IgA classification (Lee classification published in 1982, [70]), the authors observed larger numbers, however without significant differences, of M2c anti-inflammatory Mφs in glomeruli with minor lesions.

In conclusion, investigating renal biopsies from human samples, which only represent a snapshot of the current disease state, is difficult and should integrate the dynamics of Mφ subtypes in early disease and during disease progression. Thus, when we observe the association between Mφ subtypes and glomerular injury, it remains unclear whether Mφs are the cause or the consequence of these pathological lesions. Therefore, future studies are needed to address these questions.

## 5. Glomerular Macrophages in Renal Allograft Rejection

Renal transplant is considered the treatment of choice in patients with end-stage renal disease as it improves patient survival when compared to dialysis. Antigens of genetically different donors induce an immune response in the recipient, which can be potentially fatal for the graft if not addressed–this is commonly named allograft rejection. According to the histopathology and immunological characteristics, allograft rejections may be broadly classified under several categories, such as hyperacute rejection; acute antibody mediated rejection (AMR)–characterized by circulating donor-specific alloantibodies (DSA) and histopathological evidence of inflammation within glomeruli (glomerulitis) and peritubular capillary (PTC) (capillarities) [71] with cell infiltration of innate immune system including Mφ, neutrophils or NK cells -; and, acute T-cell mediated rejection (TCR) -characterized by lymphocytic infiltration of the tubule-interstitial. Chronic rejection can be both antibody or T-cell mediated [72] with chronic AMR recognized as the leading cause of graft failure [73]. Accordingly, the chemoattractant for monocytes/macrophages–MCP1–is mainly detected in urine samples of kidney transplant recipients with biopsy-proven graft rejection when compared with non-rejection transplant recipients or healthy control [74]. Therefore, Mφs, and particularly glomerular Mφs as we will see in the following sections, are fundamental players in these pathophysiological processes [75,76,77]. However, only a limited number of studies have defined their functional and phenotypic characteristics [78].

According to the literature, we can define the following points:The number of glomerular Mφs is correlated with severe allograft rejection, renal dysfunction and reduced graft survival.

Marked Mφ infiltration of the allograft has been associated with severe rejection, and glomerular Mφ infiltration in particular has been shown to be an indicator of poor graft outcome [79]. The first studies using immunohistochemical analyses were published in the 80’s and correlated the number of glomerular Mφs with severity of the rejection. Thus, Reitamo et al. characterized the spatial relationships of various inflammatory cell types to the different transplant structures in 2 cases of severe human renal allograft rejection [80]. They observed that the infiltrate around the blood vessels consisted mainly of lymphocytes, whereas the infiltrate around the tubules and within the glomerular tufts consisted mainly of mononuclear phagocytes. Hancock et al., analyzing 25 renal biopsies from 19 patients with acute cellular rejection [81], reported that in mild rejection, 32% of the tubule-interstitial infiltrating cells were T lymphocytes, of which 90% were cytotoxic-suppressor cells, and 52% were Mφs. Similarly, in moderate rejection T cells composed 42% of the infiltrate and Mφs formed 38% of the total cells. By contrast, in severe rejections, the T cell component was decreased to 15% of the cells; these were preponderantly Mφs (60%) and polymorphs (22%). Of note, in severe acute rejections, a large number of Mφs were detected in glomeruli, similar to the findings by Reitamo et al. [80]. Girlanda et al., analyzed the clinical relative impact of T cells and Mφs by correlating their presence with the magnitude of the acute change in renal function at the time of biopsy in 78 consecutive patients with histological acute rejection [82]. They found that acute allograft dysfunction was most closely related to Mφ infiltration than the T-cell infiltrate, thus implicating Mφs as a critical effector in clinical acute rejection. However, no localization of the Mφs in the different renal compartments, particularly the glomerular compartment, was reported in this study. In another study, Harry et al. analyzed a total of 50 biopsies from 42 renal transplants obtained during a 30-month period for the presence of Mφs in the glomeruli [75,83,84,85]. They observed that the prognosis for the grafts containing glomerular monocytes was significantly worse during the six months after the biopsy than for those without such cells present. Whether Mφ infiltration was an independent predictor of graft outcome has remained uncertain for a long time.

The observations of these early studies mentioned above have been confirmed and extended by subsequent investigations in the 2000s: Tinckman et al., showed that the average number of CD68^+^ cells per glomerulus represented an independent predictor of worse outcomes posttransplant following acute renal allograft rejection at a follow up of 24 months [86].

2.Glomerulitis with predominant Mφ infiltration may represent a histological marker of humoral rejection.

Glomerulitis, i.e., infiltration of inflammatory cells within glomerular capillaries, well characterizes allograft rejection. The advent of C4d staining as marker for humoral response, associated with circulating DSA, emphasized glomerulitis as one of the morphological criteria for diagnosis of AMR. Glomerulitis typically contains a mixed population of immune cells involving Mφs and T lymphocytes. Several studies analyzed the predominant cell type in the glomeruli in transplant glomerulitis according to the C4d staining. Thus, Magil et al. [87,88] reported that Mφs were the predominant cells infiltrating the glomeruli in biopsies with diffuse PTC C4d deposition, whereas lymphocyte T cells predominated in C4d negative cases.

Previous studies reported that the highest numbers of glomerular Mφ were found in cases of humoral rejection [76], confirming previously published data on the relationship between transplant glomerulitis, due to glomerular accumulation of Mφs and T lymphocytes [89,90] and AMR [91]. In contrast, Harry et al. reported no correlation between glomerular Mφ infiltration and any particular biopsy change. However, authors described a worse prognosis for the grafts containing glomerular Mφs during the six months after the biopsy [75]. Differences in population and treatments may account for the discrepancy in the results of the studies presented [86].

Fahim et al. [92] suggested that it was not the total number of inflammatory cells accumulating within capillaries of glomeruli or PTC that distinguished humoral rejection from cellular rejection, but rather the composition of the endocapillary cell population. Indeed, a predominantly monocytic cell population accumulated in C4d positive biopsies, not only within glomeruli, but also within PTC, with a median value of monocyte/T-cell ratio within PTC of 2.3 in C4d positive biopsies but only 1 (*p* = 0.0008) in C4d negative biopsies.

More recently, Bergler et al. [93] investigated infiltration into 103 allografts, reporting a notably higher prevalence of Mφ infiltration in antibody-mediated and TCR, when compared to kidneys with established IFTA. Of note, responsiveness to steroid treatment was inversely related to the prominent macrophage infiltration into the allografts. Although the severity of TCR, in terms of renal function, was linked with glomerular and perivascular macrophage infiltration, authors reported an increased CD68 infiltration in AMR (when considering only glomeruli with more than three CD68+ cells) [93]. In a previous study, Tinkman et al. described a close correlation between PTC C4d and glomerular Mφs infiltration. Moreover, a glomerular Mφs count of ≥1 per glomerulus in acute rejection resulted in an independent predictor of poor allograft function at 48 months’ posttransplant.

3.Glomerular CD68+CD163+ are predominant cells in chronic-active antibody-mediated rejection.

Chronic-active antibody-mediated rejection (c-aAMR) is histologically characterized by double contours of the glomerular basement membrane (transplant glomeropathy) and/or PTC basement membrane multilayering and evidence of concomitant or recent antibody interaction with the vascular endothelium, such as either linear C4d staining in PTC or moderate microvascular inflammation [94]. van den Bosch et al. firstly described the increased glomerular infiltration with CD68+CD163+ cells in c-aAMR compared to AMR and TMR. More recently, Sabik et al. reported that, in the context of c-aAMR, CD68+ cells are present at the mean number of four cells per glomerulus and the majority (68%) were CD68+CD163+ (2.3 per glomerulus), contrary to the tubule-interstitial compartment (39%) [95]. However, no significant association with graft function or DSA presence was found for macrophage in the glomeruli [95]. Previous findings suggest that in AMR and in c-aAMR the major pathogenetic role played by Mφs is not in the context of the pro-inflammatory function, but rather in smoldering and tissue remodeling. Differently to the c-aAMR, in AMR M2 polarization was associated with poorer graft function [96].

4.Glomerular Mφs may act through complement cascade in worsening AMR.

Humoral rejection has been identified as the main reason for the failure of kidney transplants and various antibody-mediated rejection phenotypes have been recognized. The presence of complement-activating anti-HLA donor-specific antibodies (DSAs) after transplantation is recognized as a strong determinant of kidney allograft loss [87]. The specific effects of complement-activating anti-HLA DSAs on the pathogenesis of antibody-mediated rejection have been recently analyzed by Lefaucheur et al. [97]. These authors analyzed 931 renal transplant recipients, among them 157 positive, for anti DSA. In 44/157, they identified complement activating anti-HLA antibodies by a single assay testing C1q-binding to anti-HLA DSA. Interestingly, immunostaining of the biopsies revealed extensive CD68+ Mφ infiltration in peritubular and glomerular capillaries in patients with complement-activating anti-HLA DSAs, compared with patients with noncomplement-activating anti-HLA DSAs. Moreover, the histomolecular rejection phenotype associated with complement-activating DSA was characterized by increased expression of the genes that encode chemokines CXCL11 and MIP1beta, the Fc-receptor FcgRIIIA, and macrophage tetraspanins MS4A6A and MS4A7, which is indicative of endothelial activation, IFN*γ* response, CD16-mediated natural killer cell activation, and of Mφs activation. Finally, the administration of eculizumab, an anti-C5 mAb, resulted in improvement of histo-morphological lesions, reduced Mφ infiltration and normalization of the expression of pro-inflammatory cytokines. Overall, these findings highlight a potential combined role of complement and Mφs in worsen renal lesions in AMR. Therefore, monitoring CD68^+^ cells might be important to identify patients at higher risk for AMR and early decline of graft function.

In conclusion, glomerular Mφ infiltration represents common findings in allograft rejection and were significantly associated with clinical outcome and severity, regardless of type of rejection [79,86,98,99], however, a predominant infiltration of glomerular and PTC Mφs may represent a histological marker of AMR with a predominant M2 polarization [96], that is associated with reduced outcome in terms of graft function [96].

A schematic representation of the role Mφs role in AMR is captured in Figure 3. The Mφ pathogenic role is however, still far from being completely elucidated, mainly due to their functional plasticity. More evidence on the roles and mechanistic findings of glomerular Mφs in renal graft dysfunction is fundamental in future research, offering potential clinical and therapeutic relevance.

## 6. Insights from Animal Models

This review focuses on the role of glomerular Mφs in human GNs. However, the picture in humans, as seen in the previous sections, is complicated by several dimensions such as the use of different markers to identify/characterize Mφs and Mφ subtypes and considerable differences among studies considering naïve or treated patients and length of treatment duration. Below, we have therefore considered the main insights from animal models to complement human pathology observations–namely, those with translational relevance (evidence captured in Figure 4).

Mφs in experimental GN and their role(s) in the development of glomerular lesions have been studied through four different strategies: (a) adoptive transfer experiments; (b) utilization of models of Mφs selective ablation (CD11b-DTR) under the control of Diphtheria toxin [100,101] or in mice deficient for CD11b^−/−^ (corresponding to overall target ablation, [102]); (c) reduction of glomerular Mφ recruitment via blockade of CXC16, CCL2 or MCP-1 or finally (d) use of mice deficient for complement factor H inhibitor (CfH^−/−^) to outline the complement function in relation to Mφ tissue infiltration.

With the first strategy, Ikezumi et al., using adoptive transfer studies, demonstrated that Mφs can induce proteinuria and mesangial cell proliferation [103]. Authors induced nephrotoxic nephritis (NTS) in rats, a preclinical model mimicking crescentic GN in humans. To facilitate the adoptive transfer studies, immunized animals were made leukopenic by cyclophosphamide treatment. Bone marrow-derived (BM) or NR8383 macrophages were transferred by tail vein injection 24 h after NTS, with animals killed 3 or 24 h after transfer. Compared to NTS cyclophosphamide pre-treated mice (controls), adoptive transfer led to significant glomerular accumulation of BM or NR8383 macrophages within 3 h of injection, and this was still evident 24 h later. Adoptive transfer of BM or NR8383 macrophages induced statistically increased proteinuria, glomerular cell proliferation and glomerular hypercellularity compared to NTS controls. The degree of renal injury correlated with the number of transferred glomerular Mφs. The study clearly demonstrated that Mφs, injected 24 h after disease induction can directly induce renal injury in terms of significant proteinuria and mesangial cell proliferation. The rigor of this approach is shown by the highly significant correlation between the number of transferred glomerular Mφs and the severity of renal injury. The same team at Monash University, in a subsequent work, demonstrated that Mφ stimulation with IFN-γ significantly augmented macrophage-mediated renal injury in NTS when Mφs were injected 1 day post disease induction [104]. Notably, the authors observed an increase in the number of glomerular Mφs 24 h after transfer due to both an increase in glomerular Mφ recruitment at 3 h after transfer and an increased retention of Mφs within the glomerulus up to 24 h after transfer [104]. Similarly induction of NTS in CD11b-DTR mice showed that Mφ ablation between day 15 and day 20 after NTS induction, a phase of the disease characterized by progressive disease, attenuates glomerular Mφ infiltration [105]. Similarly, reduction of Mφ infiltration by blockade of Mφ chemo-attractive chemokines was comparably effective in NTS [106] and in Wistar rats developing antibody-mediated anti-glomerular basement membrane GN [107]. Finally, CfH^−/−^ mice developed severe NTS with diffuse glomerular proliferation and crescents that were associated with accumulation of Mφs whereas CD11b^−/−^ mice were indistinguishable from wild-type mice [102]. Similarly, renal adoptive transfer of Mφs genetically modified to express either IL-4 [108], IL-10 [109] reduced Mφ infiltration and renal injury in animal nephritis models.

Overall, those preclinical observations in NTS, suggest that (a) selective ablation or reduced Mφ infiltration results in reduced glomerular lesions; (b) stimulation of Mφ with IFN-γ, therefore resulting in a a M1 phenotype, significantly augmented Mφ-mediated renal injury, (c) complement is a co-player with Mφs in the sequence that leads irreversible renal damage in NTS, and (d) stimulation of cytokines IL-4 and IL-10 expression by Mφs, both cytokines leading respectively to the subtypes M2a and M2c, was protective.

In other preclinical model of glomerulopathy resembling focal segmental glomerulosclerosis in humans such as the adriamycin nephropathy (AN), a rodent model initiated by podocyte injury [110], partial Mφ depletion using monoclonal antibody directed against CD11b/CD18 integrin started before (but not after) AN protected both renal function and structure [111]. The same group also showed that severe combined immunodeficient mice infused with M1 macrophages had a more severe histological and functional injury, whereas M2 macrophage-induced transfused mice had reduced histological and functional injury [112].

Similarly, in a recent manuscript, Wei et al. [113], demonstrated in a rodent model of acute AMR that MHC I DSA upregulated genes related to monocyte transmigration promoting monocyte differentiation into CD68+CD206+CD163+Mφ, enforcing the concept that infiltrating Mφ may contribute to vascular injury during AMR.

## 7. Conclusions and Future Directions

We have gathered here the entire preclinical and clinical literature referring to the role of macrophages in auto- and allo-immune nephritis. The literature in human is sparse and knowledge relies on old immunohistochemical studies using several cell markers characterizing the different monocytic/Mφ cell populations. Data in GNs relies on a few hundred cases characterized using CD68, a marker unsystematically adopted by pathologists in routine clinical practice. In chronic allograft rejection, more data are available and there is now a consolidated body of evidence indicating that Mφs play a key role as determinants of graft glomerulitis, and are involved in renal graft loss. Evidence from preclinical models indicates that as a cell type, Mφs are able to profoundly modulate disease progression. Generally, Mφ depletion, during disease onset, results in a reduced glomerular lesion, evidence corroborated by better outcome in patients with lower number of glomerular Mφs. Mφ polarization toward specific sub phenotypes obviously complicates the role of Mφ in auto- and allo-immune nephritis, though consistent evidence exists in both humans and rodent models for a more profound detrimental role of M1 compared to M2 sub phenotype. More generally, literature in both humans and preclinical models suffers from significant bias; in humans it is primarily related to the very limited biopsy sample size, while in preclinical models it lies mainly in the experimental setup with Mφ depletion performed prior to, or in the immediate hours following disease induction. The availability of specific biomarkers of Mφ activation, rather than simply labelling Mφ presence in tissue, would also dramatically enhance the ability to assess the role of these cells in the context of GN and renal transplantation. This could potentially enable improved monitoring of drug activity and ultimately the development of novel therapeutic strategies to target this, still very elusive, cellular player.

## Figures and Tables

**Figure 1 cells-10-00603-f001:**
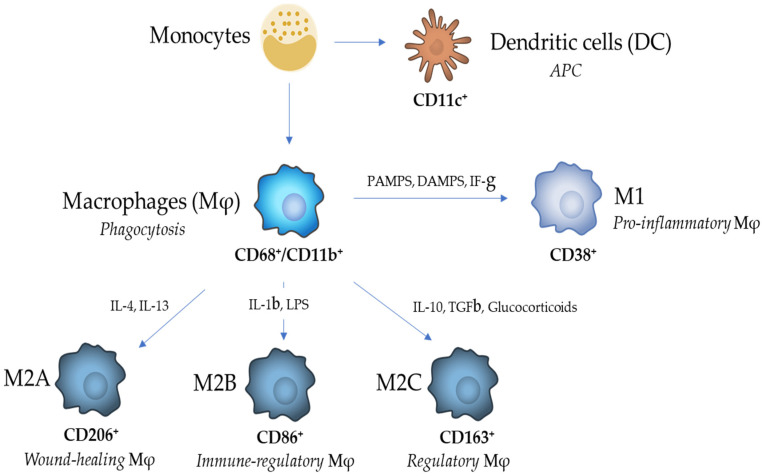
Schematic representation of the monocytic/macrophage differentiation lineage with related markers and cellular functions.

**Figure 2 cells-10-00603-f002:**
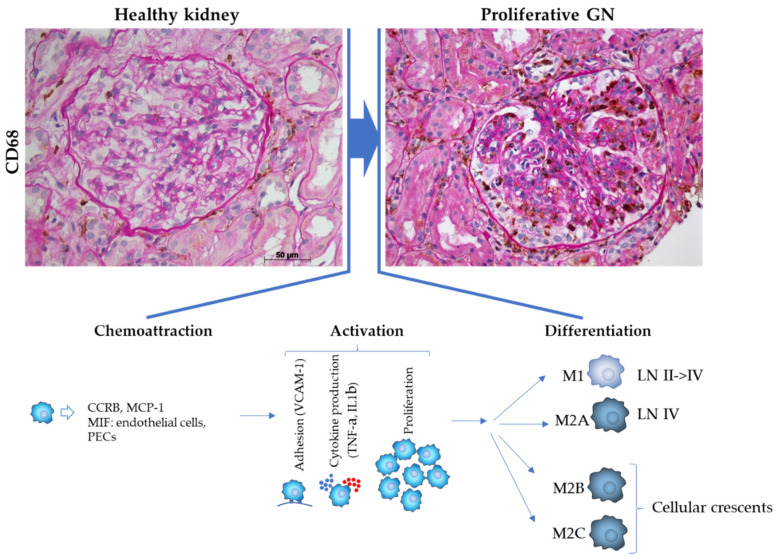
Representative images of healthy and proliferative glomeruli and schematic representation of pathophysiological processes involved in monocyte/macrophages transition occurring during disease onset and progression. Histological images labeled for CD68 (macrophages, brown) and counterstained with periodic acid-Schiff (PAS) staining (bar = 50 µM).

**Figure 3 cells-10-00603-f003:**
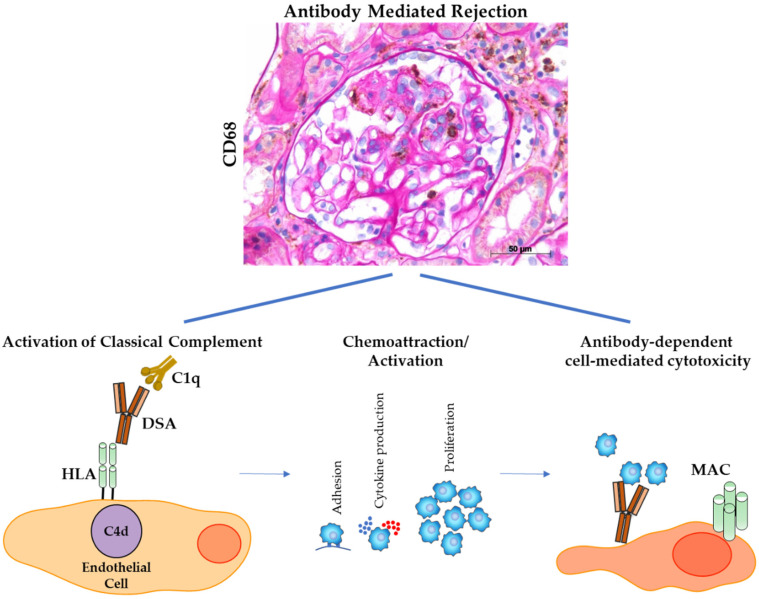
Representative image of C4d+ glomerulus of renal transplant and schematic representation of potential combined role of complement and Mφs in worsen renal lesions in Antibody Mediated Rejection. Histological image labeled for CD68 (macrophages, brown) and counterstained with periodic acid-Schiff (PAS) staining (bar = 50 µM).

**Figure 4 cells-10-00603-f004:**
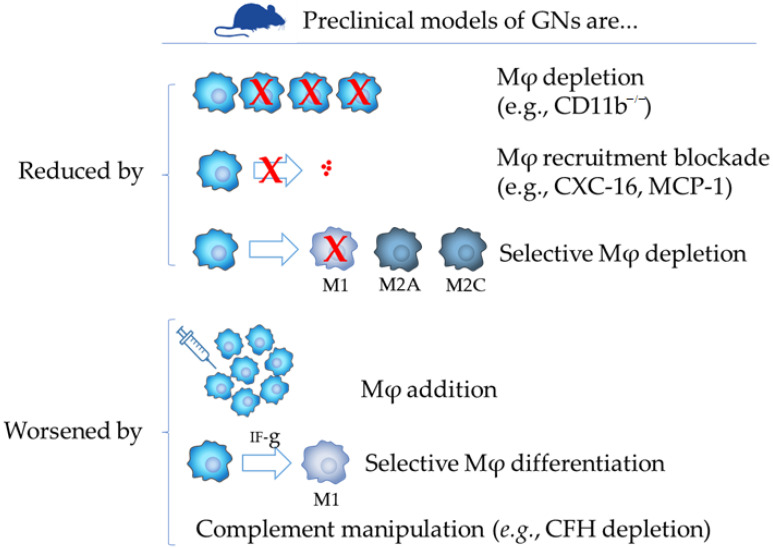
Schematic representation of scientific evidence supporting a role of monocytic/macrophages in experimental model of glomerulonephritis.

**Table 1 cells-10-00603-t001:** PubMed list of publications and total number of patients studied reviewed in the present manuscript.

Glomerular Pathology	Glomerular Macrophages Phenotypic Marker CD68	Referefnce	Glomerular Macrophages Phenotypic Marker LeuM3 ^1^, FM32 ^2^, Esterase ^3^, EDI ^4^	Reference
Post-infectious Gn	-		27 (intense ^3^)	[26]
IgA Gn	2 (positive)	[31]	35 (negative ^2^)	[29]
			5 (intense ^3^)	[26]
			8 (positive ^3^)	[27]
Membranous Gn	6 (positive)	[31]	-	
	8 (intense)	[31]		
RPGN	20 (intense)	[28]	20 (positive ^1^)	[28]
			20 (intense ^3^)	[26]
S Henoch GN	-		8 (intense ^1^)	[27]
Lupus Nephritis	3 (positive)	[31]	61 (intense ^3^)	[27]
	17 (positive)	[33]	7 (intense ^1^)	[27]
	8 (intense)	[33]		
ANCA vasculitis	1 (positive)	[31]	6 (intense ^1^)	[27]
	25 (intense)	[32]		
	8 (intense)	[32]		
Cryoglobulinemic Gn	-		29 (intense ^3^)	[26]
Anti-GBM Gn	-		3 (intense ^1^)	[27]
Cyclosporin Toxicity	-		6 (positive)	[30]

## Data Availability

Not applicable.

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
