# Peer review of "Glomerular Macrophages in Human Auto- and Allo-Immune Nephritis"

_cells, 2021, doi:10.3390/cells10030603_

Round 1
Reviewer 1 Report
The manuscript titled “Glomerular macrophages in human auto- and allo-immune nephritis” is a thorough and well-researched review. As opposed to most other reviews on renal mononuclear cells that focus on the interstitium, the authors decide to focus on the glomerular space. It provides a nice explanation of the gaps of knowledge in the field of autoimmune and alloimmune kidney disease.
Suggestions with potential to significantly increase the impact of the manuscript:
[1] Page 4, lines 176-188: The authors nicely discuss studies using immunostaining to compare numbers of macrophage between different native glomerular diseases. But it might be helpful to the reader to know the contemporary “best practices” for enumerating glomerular macrophage infiltration. Are their recommended methods that are more rigorous or reproducible? Has reproducibility of glomerular macrophage staining/scoring between pathologists or between laboratories been published?
[2] Although it is not yet known whether distinct chemoattractants explain the differences in glomerular macrophage localization between necrotizing GN, crescentic GN, and non-necrotizing GN, this is left unstated in section 3 on page 5 (lines 201-225). Do authors agree that it is unlikely the chemoattractant that is specific to the glomerular compartment, but rather the inducer of chemoattractant production? This could have implications for pharmaceutical development.
[3] The discussion of macrophage adhesion molecules such as VCAM (page 5, lines 226-248) does not include the many publications that find urinary soluble VCAM1 levels that correspond with disease activity, such as lupus nephritis. Could it be possible that cleavage of VCAM1 and/or other adhesion molecules after diapedesis may explain the lack of expression in some glomerular macrophage? The same question arises when reading about soluble CD163 on page 7;lines 315-319: do CD163- macrophage arise from CD163+ cells that have undergone cleavage events in situ?
[4] For discussion of macrophages in allograft rejection, authors should be clear when they are discussing acute cellular rejection with only tubulitis, acute cellular rejection with arteritis, hyperacute rejection, acute antibody mediated rejection with only capillaritis, AMR with glomerulitis, transplant glomerulopathy, or chronic AMR. It seems likely that macrophage would play different roles in each of these pathologies. The literature is complicated by our evolving understanding of vascular and antibody mediated rejection, and this significantly confounds many published human transplant studies.
Minor issues / suggestions:
[5] page 3, 103-111: description of M2 macrophage subtypes appears to be incomplete. It is unclear how M2c differs from M2a or M2b, and unclear how cell surface markers have been used to distinguish. Only a single reference is provided for each subtype. Additional references and would be appreciated. Given the caveats presented in the subsequent paragraphs, it would be important to know if this classification system has been validated by different laboratories and in different species. Maybe we should just think of glomerular CD163+CD68+ macrophage as an intermediate stage of differentiation between anti-inflammatory and pro-fibrotic.
[6] page 3, line 142: “whereas those infiltrating the glomeruli were DC-SIGN” should this be “negative for DC-SIGN”?
[7] formatting error for TNF-alpha (page 5, lines 237 and 240).
[8] Unfortunately, the studies of macrophage subsets in IgAN were probably underpowered (page 7, lines 320-335). So lack of significant findings in IgAN patient subgroups based on MEST-C scoring might be real, or might be due to insufficient patients studied.
[9] typo page 8, line 357, capillaritis
[10] consistency of nomenclature, MCP-1 is used to discuss GN but CCL2 is used to discuss transplant rejection (page 8, line 361)
[11] for study by Hancock et al. (page 8, line 378-385), are the proportions noted indicative of glomerular infiltrates, or tubulointerstitial?
[12] page 9, lines 424-429: The genotyping study by Chen et al. look at MIF polymorphisms in the recipient blood cells, not the donor kidneys. So the argument presumed by the author that parenchymal glomerular cells produce MIF to mediate glomerular macrophage infiltration was not tested in this study. More likely, the MIF and BAFF polymorphisms studies by Chen et al. influenced loss of tolerance for donor HLA and/or development of DSA.
[13] page 9, line : instead of listing genes identified as upregulated in complement-activating DSA (CXCL11, CCL4, MS4A7, MS4A6A, and FCGR3A), it might be better to write “the genes that encode chemokines CXCL11 and MIP1beta, the Fc-receptor FcgRIIIA, and macrophage tetraspanins MS4A6A and MS4A7.”
[14] page 10, line 497, use of glomerulitis to describe disease in nephrotoxic serum nephritis animal model is not appropriate. Glomerular macrophage infiltration would be a better phrase.
[15] Figure 1, typo, should change to “PAMPS, DAMPS, IFNgamma”
Reviewer 2 Report
I read with interest this manuscript. This review discuss the role of glomerular macrophages in human auto- and allo-immune nephritis.
Patients after transplantation may have acute antibody-mediated rejection or cell-mediated rejection . The presence of anti-HLA Abs against donor is a major cause of acute antibody-mediated rejection developing, in contrast to cell-mediated rejection, where mainly T-cells are the mediators of this type of rejection. The authors should describe in more detail the role of glomerular macrophages in both types of graft rejection as well as in chronic allograft nephropathy.
In conclusions the authors should discuss what are the key findings and weaknesses in the research done in this field so far.
